# Non-Destructive Characterization of Industrial Membrane Cartridges by Using Liquid–Liquid Displacement Porosimetry (LLDP)

**DOI:** 10.3390/membranes10120369

**Published:** 2020-11-25

**Authors:** René I. Peinador, Mohamed Kaabouch, Roger Ben Aim, José I. Calvo

**Affiliations:** 1Laboratoire Mesures et Charactérisations, Institut de la Filtration et des Techniques Séparatives (IFTS), Rue Marcel Pagnol, 47510 Foulayronnes, France; rene.peinador@ifts-sls.com (R.I.P.); roger.ben.aim@ifts-sls.com (R.B.A.); 2Département de Mécanique, Faculté des Sciences et d’Ingénierie (FSI), University of Toulouse III-Paul Sabatier, 31400 Toulouse, France; mohamed.kaabouch@univ-tlse3.fr; 3Departamento de Física Aplicada, ETSIIAA, Universidad de Valladolid, 34071 Palencia, Spain; 4Institute of Sustainable Processes (ISP), Dr. Mergelina s/n, 47071 Valladolid, Spain

**Keywords:** membrane characterization, pore size distribution (PSD), filtration cartridges, LLDP, GLDP

## Abstract

This works aims to propose and demonstrate the accuracy of a novel method of characterization aimed for non-destructive analysis of microfiltration (MF) membrane cartridges. The method adapts conventional liquid–liquid displacement porosimetry (LLDP) for performing an in-line porosimetric analysis of the membrane cartridges, getting their pore size distributions (PSDs) and mean pore diameters (*d_avg_*). Six commercial filtration cartridges featuring polyethersulfone (PES) pleated membranes were analyzed using a newly designed filtration rig, based on the liquid–liquid displacement porometer, developed at the Institut de la Filtration et des Techniques Séparatives (IFTS) and operated at constant flow. The experimental rig allows the direct and non-destructive characterization of the cartridge in its original presentation. Results have been compared with those obtained by using gas–liquid displacement porosimetry (GLDP) on small membrane coupons detached from such cartridges. The comparison allows us to conclude that the proposed method gives enough accuracy in the determination of porosimetric characteristics of the filters. This method can be used as a precise characterization technique for a non-destructive in-line study of filter performance and can be envisaged as useful to periodic quality or fouling control of the commercial cartridges.

## 1. Introduction

Microfiltration (MF) is a membrane process developed in the early 20th century. In fact, MF is, after dialysis, the first process developed in what is currently known as membrane technology. Following Traube’s pioneering work, Zsigmondy was the first to develop a cellulose nitrate membrane suitable for the sterilization of small amounts of fluid [1]. Sartorius commercialized for the first time MF membrane cartridges [2].

Still today, after nearly a century of continuous development of membrane technology, sterilization is one of the most widely used applications of MF. The use of a 0.22–0.2 μm MF membrane before bottling is a recommended standard for avoiding the development of bacteria in water [3]. Nowadays, applications of MF cover a broad range of processes in the water reuse, pharmaceutical, juice, and beverages industries.

According to the safety regulations involved in the sterilization of food, beverage, or pharmaceutical products, the membranes intended for such uses have to fulfill the more and more strict requirements. In the beginning, a simple 0.45 μm was supposed to be enough to assure bacteria sterilization. Nevertheless, soon it was discovered that *Pseudomonas diminuta* and similar small-size microorganisms were able to cross the membrane and to break the assumption of complete sterilization. Then, filters of smaller pore size needed to be used. This is why it is usual now to consider 0.45 μm filters as clarification filters and 0.2 μm filters (even 0.1 μm, if necessary) as sterilization filters. In the case of MF filters designed for liquid filtration, the performance of these filters must be tested in accordance with ASTM F848-15a standard [4], based on the retention of *Brevundimonas diminuta* as the challenge organism.

On the other side, if the initial applications of MF were concerned with separations in small amounts of fluid (often at lab scale), the widespread use of membranes in all kind of industrial processes soon led to the necessity of increasing the volume and the flow rate of fluid to be treated. This led to the design of compact cartridges using flat sheet membranes (large membrane area for a small cartridge volume).

Moreover, in the case of MF applied for water sterilization, the filter should eliminate a low concentration of suspended solids. The common features of large volumes to be treated and low concentrations to be removed pushed membrane manufacturers to develop the pleated membrane cartridge as one among possible convenient solutions [5]. Pleating of microfiltration membranes has become the most common feature of cartridges used in a range of biotechnology applications today [6]. A pleated cartridge consists of a flat sheet microfiltration membrane many times folded and packed around a central core surrounding a central perforated core (the output channel) [7] in a way that pleating density gets maximum value [5]. On the other side, the adequate flow pattern is assured by appropriated spacers to hold pleats open [8].

Additionally, a proper evaluation of given filter characteristics usually starts by using small membrane discs to assess the performance of the membrane. The question is, how reliable is the information obtained from such characterization when applied to scaled filters which also have a more complex configuration [9]? For example, it has been reported clean water fluxes in small (47 mm) membrane discs double those later obtained for commercial pleated cartridges [5], and even higher discrepancies appear in high fouling conditions [10]. Moreover, the heterogeneity of the porous structure of the membrane has also been reported [11]: the characteristics of several discs taken from the same membrane sheet are quite different when measuring permeability, porosity, and pore size. Obviously, a direct characterization of the commercialized product (cartridge) should solve such discrepancies. However, this is not a simple task; for technical reasons, the pore size characterization is generally done on small membrane samples—our goal was to do it directly on the commercial cartridge.

Characterization of membrane materials is necessary knowledge to decide the appropriate application in which such membranes will be successfully applied. This characterization comprises the determination of a number of parameters, among which can be included pore size, molecular weight cut-off (MWCO), fractional efficiency, porosity, thickness, permeability, hydrophobicity/hydrophilicity, adsorption, crystallization, tensile and mechanical properties, etc. In terms of pore size, the most complete information comes from determination of the (average) pore size diameter and the pore size distribution (PSD) [12]. There are many characterization methods able to supply the PSD of a membrane, and those can be classified into two main groups: direct [13,14] and indirect [15,16,17,18] methods [19].

Usually, MF membranes are characterized in terms of mean pore size (in fact, pore diameter) or PSD [20]. For such membranes, the mostly used characterization method (and recommended standard [18]) is the gas–liquid displacement porosimetry (GLDP), also known as capillary flow porometry (CFP) [12].

Applying GLDP to large membrane modules like those concerned here is a difficult task, as large flowrates of air are involved in the measurement (involving the use of big and noisy air compressors) and uncertainty associated with the bubble-point-based methods increases when large membrane areas are involved [21,22]. Certainly, it is always possible to open the cartridge, cut some pieces of the membranes pleated inside, and analyze them in a usual flat disk cell (47-mm diameter) common to most commercialized gas porometers. However, as previously commented, this could result in differences with the actual PSD of the cartridge due to slight changes and compressions in the membrane material during the pleating process. On the other side, using a liquid as displacing fluid, instead of air, should reduce greatly the fluxes involved, making an accurate control of them easier. For this reason, the authors decided to explore the possibility of using liquid–liquid displacement porosimetry (LLDP) instead of GLDP for analyzing such membrane modules.

Another derivation of the original GLDP characterization method, also proposed by Bechhold [23], is the well-known liquid–liquid displacement method, whose main difference with GLDP relies on the use of a pair of immiscible liquids for wetting and displacing instead of air. The major advantage of LLDP over GLDP is the lower surface tension of the interface liquid–liquid membrane as compared with the air–liquid membrane one, which makes LLDP suitable to analyze membranes having pores smaller than the usual MF range. In fact, LLDP is one of the most accurate available methods to determine PSD for membranes aimed for ultrafiltration (UF), or even nanofiltration (NF) [24,25,26,27,28].

In this work, a novel method derived from conventional LLDP has been developed and tested to characterize several commercial (MF or disinfection) filters marketed for water bottling filtration. Six membrane cartridges from different manufacturers and mean pore sizes were analyzed by using LLDP, and results have been compared with the PSD obtained from GLDP applied to coupons cut from the inner membrane of such cartridges. This new LLDP-based method of characterization is able to obtain the porous information of industrial membrane cartridges. The main advantage of the method is that no cartridge destruction is needed to analyze the membrane pieces inside the cartridge. To our best knowledge, no previous paper has been published where commercial membrane cartridges were studied to assess their porosity, mean pore size, or pore size distribution. Usually, membrane small coupons detached from the cartridge or from a flat sheet supplied apart by the manufacturer were used in such studies. Moreover, the technique could be adapted for regular checking of the changes in membrane performance due to fouling or particle deposition.

## 2. Fluid Displacement Porometry (FDP) Principles

The fluid–fluid displacement porosimetry technique, also known as capillary flow porometry (CFP), comprises two similar techniques, gas–liquid displacement porosimetry (GLDP) and liquid–liquid displacement porosimetry (LLDP), both derived from the original determination of bubble point for membrane filters.

The bubble point method and all porosimetric techniques derived from it are based on the well-known Young–Laplace (Equation (1)) equation which governs the pressure difference at the interface between two immiscible fluids [19].

From an operation point of view, both techniques consist of pressurizing a fluid (gas or liquid), which is then forced to enter inside the pores of a filter which has been previously wetted by a proper wetting liquid. The procedure consists of steeply increasing the applied pressure in such a way that pores of decreasing sizes are opened in each step. When this fluid pressure overcomes the capillary intrusion pressure of the largest pore, the displacement fluid can penetrate into those biggest pores pushing out the wetting liquid of them and then flowing through the membrane. The pore size *d_p_* (m) of the pores opened to flow at each applied pressure, Δ*p* (Pa), is calculated by the Young–Laplace equation:(1)$$\Delta p=\;\frac{4\gamma \cos\theta }{d_{p}}$$
where *γ* (N/m) is the surface tension of wetting liquid and *θ* the contact angle between the liquid and capillary wall. To get reliable results, the value of cos*θ* must be assumed equal to 1 (perfect wetting conditions) for both techniques. The pressure at which the displacement fluid starts to flow through the pore is called a bubble point (GLDP) or droplet point (LLDP) and the pore size at the bubble/droplet point (according to Equation (1)) is matched with the maximum pore size of the filter. When the pressure is further increased, the displacement fluid flows through smaller pores and flux of the displacing fluid can be measured.

Once a complete flux–pressure run has finished, it is easy to obtain the contribution of each class of opened pores to the total permeability (defined as the flux to pressure ratio) from the next equation:(2)$$\Delta L_{k}=\frac{L_{k}-L_{k-1}\;}{L_{tot}}$$

*L_k_* is the permeability of the *k*-th experimental step (*k* = *1*, *2*, *…*, *i*) and *L_tot_* is the final permeability (asymptotic permeability) in the final step (i.e., *k* = *i*). The value of the asymptotic permeability, *L_tot_*_,_ corresponds to the moment when the wetting liquid is drained away from all the membrane pores. The plot of the percentage of the total permeability achieved in each pore size opened can be understood as a pore size distribution in terms of flow (or permeability).

To convert this permeability distribution into the absolute number of pores opened at each experimental step, a suited model for fluid flow inside the opened pores is needed. For instance, in the case of GLDP, the intruding fluid is a gas whose transport model could be convective (Hagen–Poiseuille) or molecular (Knudsen) depending on the relation between gas molecules’ mean free path and pore size. In the case of LLDP, only convective flow must be considered. Finally, a proper modeling of the experimental results should assume a given model for the porous structure: the model generally selected of parallel not interconnected network of cylindrical pores may be far from the real structure. In any case, permeability-based PSD, as coming from direct experimental data with no modeling, can be considered a good approach to characterize the actual structure of the studied filter.

## 3. Materials and Methods

Six membrane filter cartridges (10″) supplied from recognized manufacturers were characterized. All of them have a similar pleated structure and use a flat polyethersulfone (PES) membrane, offering a broad chemical compatibility, exceptional high flow rates, and high total throughputs. The cartridge filters were studied under a non-disclosure agreement that prevents the publishing of the company or product name (these will be designated as Companies A to D respectively).

The information available on the membranes as supplied by manufacturers is summarized in Table 1.

For the LLDP analysis (non-destructive method), cartridges were placed in a specifically designed rig using standard. After LLDP analysis, the same cartridge was characterized by GLDP (destructive method). For that purpose, the sheet of pleated filter media was unfolded (see Figure 1b) and several coupons of this sheet were cut in order to be tested in a specific 25 mm diameter cell installed on the Institut de la Filtration et des Techniques Séparatives (IFTS) porometer. PES MF filters present a typical sponge-like structure as can be seen in Figure 1a.

It is obvious, from Figure 1a, that determination of the pore sizes of such a membrane from any visual inspection of the pores, even aided by any of the available image analysis software, is subjected to a great margin of error. As can be seen, pores with apparent sizes as variable as 618 to 276 nm can be distinguished in this picture. Therefore, a more precise and reliable characterization method should account for the actual PSD of these membranes. In our case, GLDP and LLDP have been used for such characterization.

### 3.1. GLDP Characterization

The gas–liquid porometry analysis was performed with an IFTS fluid–fluid porometer (FFP) (model IFTS-PRM-8710^®^, Foulayronnes, France) consisting of an automated pressure constant device suitable for working in gas/liquid and liquid/liquid configurations, both at lab scale. The device, when working in GLDP mode, allows pore size measurement in the range of 64 nm to 200 µm. This equipment was designed for getting very stable pressure of clean air (accuracy ± 0.1 mbar) and very accurate measurement of resulting fluxes by means of a mass flowmeter (accuracy ± 1 mL/min). The original software is able to determine several important parameters related to the raw data obtained, including mean pore diameter, peak pore size, PSD, fluid permeability, and bubble point (maximum pore size in the sample). The equipment is easily adapted to several membrane configurations or modules, including hollow fiber, tubular, and flat sheet.

For GLDP runs, the filter media must be previously wet in an appropriated liquid (wetting phase). Perfluoro halogenated compounds are the most successful liquids that meet requirements of complete wettability, low surface tension, low vapor pressures, and low reactivity [19], and accordingly, most GLDP manufacturers usually supply these liquids to their customers (e.g., Porofil^®^, Silwick^®^, Porewick^®^, Galwick^®^, with the latest based on different configurations of Fluorinert^®^) [12]. In this study, the wetting liquid was fluorocarboned commercial liquid Fluorinert FC-43 (3M^®^), having a surface tension value of 16 mN/m. Three membrane coupons cut from different locations of the unfolded cartridge sheet were gently soaked in the wetting liquid for half an hour before being put in the measurement cell. Results of GLDP analysis for each coupon of the different cartridges were averaged and experimental errors calculated as standard deviation.

### 3.2. LLDP Characterization

For the case of LLDP analysis of industrial cartridges, the IFTS porosimeter used for GLDP was not able to manage the high fluxes involved in the analysis of an original cartridge. Therefore, an industrial cartridge adapted version of the porosimeter was developed and built specifically for this purpose. The device consists of an experimental single pass test system, manually controlled, schematically shown in Figure 2.

Main elements of the porosimeter set up are briefly described:(1)Stainless steel tank with 20 L of capacity used to collect the porosimetric phases prior to be pumped through the filter cartridge.(2)Centrifugal pump Lowara, mod. CA70, from Xylem^®^, New York, USA. It has a flow range from 1 up to 80 L/min allowing precise and steeply control of the flux implemented across the cartridge filter (accuracy: 1% of set point).(3)Bypass valve used for mixing both porosimetric liquids.(4)Thermostatic bath with regulator to control the temperature of the testing liquids.(5)Mass flowmeter from Emerson^®^, micromotion mod R-series, Ferguson, USA: its range goes from 0.1 L/min to 25 L/min (accuracy of 1% of set point).(6)Microfiltration (Pall^®^ Fluorodyne 0.45 µm, New York, USA) pollution filter in which aqueous phase is prefiltered before being pumped through the filter cartridge. This ensures the depollution of the fluid introduced in the rig.(7)Temperature controller (Barber-Coleman ®, type 2204E, Illinois, USA) connected to the filter housing input (operation range: 0–200 °C, temperature accuracy: 0.1 °C).(8)Differential piezo resistive pressure transducer (Keller-druck^®^, type PR-33K, Winterthur, Switzerland) connected to the filter housing input (operation range: 0–10 bars, pressure accuracy: 0.001 bar).(9)Filter housing with diameters from 63–70 mm made of stainless steel and adapted to both industrial and sanitary grades.(10)Flow regulator valve conditioning the permeate flow during purge protocol.

Measurement sensors (mass flowmeters and pressure transducer) are connected to a PC through serial ports (RS-232C). Implemented software Microsoft^®^ Visual studio 2019, Net framework includes data acquisition and treatment.

Similar to GLDP, wetting and displacement liquids used in LLDP analysis are to be carefully selected according to the membrane hydrophobicity/hydrophilicity and good chemical compatibility with the material matrix active layer and with the porometer internal parts to avoid corrosion or degradation. Wetting fluid should also exhibit appropriated physical properties as low displacement viscosity and low vapor pressure. A very stable 7:2 (*v*/*v*) binary mixture composed of water/isobutanol (γ =1.7 mN/m) was used in this work. This mixture was prepared by mixing proper amounts of Milli-Q^®^ grade water (~14 L) and isobutanol (Methyl-2-Propanol) (~4 L) into a reservoir tank mixing both phases using variable frequency pump connecting the bypass valve for 60 min at half frequency rate. After mixing for getting the respective saturation of the two phases, a settling period of 24 h results in a clear separation of the two phases. The lighter density, alcoholic-rich (organic) phase is firstly drained off (by an aspiration head on pump) and used as wetting phase while the aqueous phase (remaining in the lower part of the reservoir tank due to its higher density) is used as the displacement one.

Prior to LLDP characterization, cartridge filters were dried for 24 h at 45 °C in a stove (after the previous permeability determination, as explained in next section). The cartridge was then installed in its housing (9) and it was filled with the wetting phase (the alcohol-rich phase of the demixing) for half an hour at atmospheric pressure. Then, cartridge housing was closed, and tank reservoir was filled by aqueous displacement phase, being system purged at a constant flow of 1 L/min for some minutes, to avoid bubble entrapment. The experiment started with a small initial flux imposed on the pump: this value is settled between 0.1 to 1 L/min.

Data pairs (flow imposed and pressure reached) were recorded at each flux step. In fact, pressure varied slowly before stabilization. This evolution was recorded and the slope of the curve pressure vs. time was calculated. When this slope reached a value of 0.1 to 1 mbar/s, we considered that equilibrium was reached.

The experience was finalized when the resulting S-shaped curve reached the line corresponding to the permeability measurement, meaning that all the pores were opened to flow so that the permeability to the displacing liquid became constant, as can be clearly seen in Figure 4 (left). Usually it took 2–2.5 h to finish a complete run.

To increase the certainty of results, each cartridge was analyzed two times following the previously described procedure, and resulting parameters were averaged.

### 3.3. Water Permeability Measurements

To check if LLDP analysis affected the performance of the cartridges, a simple measurement of water permeability for these filters was performed before and after the LLDP analysis.

For such purpose, the reservoir tank (1) in Figure 2 was filled with ultrapure water, then the filter cartridge was installed into the housing cell (after being carefully dried) and pure water was pumped through the filter initially at a constant flow of 1 L/min to remove air bubbles trapped. Finally, water permeability was obtained from the slope of 50 data points (pressure, flow) obtained by consecutive steps of ~0.5 L/min until 25 L/m and averaging the pressure data, once equilibrium reached, at each step.

Water permeability determination was done again after finishing LLDP analysis (previously used filter was cleaned with pure water and dried again for 24 h) and both permeability values compared.

## 4. Results and Discussion

### Comparison of Pore-Size Distributions (PSDs)

An example of the porosimetric GLDP run for one coupon obtained from the Company C cartridge is shown in Figure 3 along with the resulting PSD in terms of contribution of each pore to the membrane permeability.

It could be a bit surprising to find that PSD ranges from 0.2 μm to 0.4 μm, with the most probable value around 0.3 μm. Apparently, this membrane must be termed most properly as a 0.3 μm filter. Nevertheless, it must be remembered the application of these kinds of filters. Since they are intended for sterilization of bottled water, the objective must be to assure no bacteria or pathogens should pass through them. Accordingly, when a 0.2 μm filter is selected, the customer is usually looking for a filter assuring that no particle larger than this size could be allowed to permeate this filter. Then, as shown in Figure 3, cartridge C fulfils this requirement. Concerning this assertion, the submicron filtration efficiency was determined according to internal IFTS protocol (IFTS_FEEIS_01 2013) using as test fluid ultra-pure water and contaminant several NIST^®^ traceable microspheres of latex with mean diameters from 0.1 to 1 µm. The counting of the particles (PMS, LiquiLaz^®^ S02, Colorado, USA) online both up and downstream the filter showed ~99% of filtration efficiency for 0.2 µm and 100% starting from 0.3 µm up to 1µm for the whole cartridges analyzed.

Similarly, Figure 4 shows the porosimetric LLDP run and resulting PSD for C cartridge.

In the case of LLDP, application of the Hagen–Poiseuille model for convective transport of liquid inside the pores can be used to convert the pore size distribution in terms of permeability to a new PSD based on number of pores in each size class. This kind of distribution is shown in Figure 5 for the selected case of Company *C*.

Similar to what happened with GLDP results, PSDs are centered in values clearly higher than the nominal 0.2 μm value.

From curves similar to those shown in Figure 3, Figure 4 and Figure 5 for the rest of the cartridges analyzed, several parameters were obtained. In the case of GLDP, the PSD is characterized in terms of the average pore size (*d_p_*_, *avg*_) along with the mode of such distribution (*d_p_*_, *mode*_). While for LLDP characterization of the cartridges, two values have been selected, the average of both distributions, the permeability-based distribution (*d_p_*_, *avg*, *fl*_) and the pore number one (*d_p_*_, *avg*, *nr*_).

These parameters for all six cartridges analyzed are presented in Table 2.

Finally, Figure 6 shows a comparison of the resulting parameters along with their experimental errors, for a clearer view.

Firstly, it must be pointed out the good reproducibility obtained for the analysis of all cartridges here studied. This results in low dispersion of measured parameters (around 10% for LLDP and even lower for GLDP). This reproducibility accounts for the regularity of the manufacturing process (coupons detached from different parts of the filter lead to very similar results) but also for the accuracy of both GLDP and LLDP methods. Certainly, GLDP gets lower dispersion which makes this technique very suitable for analysis of MF membranes (in fact, it could be considered the standard method for such filters’ characterization [29]). LLDP results also present a reasonability low dispersion, even working with big fluid amounts, which causes the measurement to necessarily have lower precision.

Regarding the comparison of GLDP and LLDP results, this comparison is mostly good, with GLDP average value quite close to the mean pore size obtained from LLDP in terms of flow (*d_p_*_, *avg*, *fl*_). This comparison is not good for the case of number-of-pores-based PSD (*d_p_*_, *avg*, *nr*_). This is quite normal as number of pores distribution is based on several assumptions about the transport mechanism acting through the pores, while *d_p_*_, *avg*, *fl*_ comes directly from experimental data. Otherwise, the mean pore size quoted for GLDP, *d_p_*_, *avg*_, is also a value corresponding to permeability direct data.

Only for some of the cartridges, namely A and D1, can a remarkable difference be found (mostly for D1 as cartridge A also matches averaged values). Additionally, cartridge D1 presents the highest experimental errors for LLDP measurements.

In general, LLDP-averaged pore diameters based on pore number show smaller values than GLPD-averaged ones and are much closer to the nominal diameters announced by the manufacturer.

Different remarks can be considered to explain these differences, concerning the quite different values of interfacial tension involved in both techniques (16 mN/m for GLDP and 1.7 mN/m for LLDP) which result in lower pressure values involved in LLDP measurements. This, if desirable as it reduces the risk of membrane distortion or breaking, also leads to higher uncertainty in the measurement of equilibrium pressures. In the case of LLDP, a minimal pressure required to get the biggest pores or “droplet point” is point out between 0.05 to 0.150 bar for whole cartridge being up to ~1.1 bar for bubble point in GLDP experiences where a possible distortion of the matrix active layer due to high pressures could shift the PSD to bigger values affecting structural parameters.

Another key parameter used in this work is related to surface filtration: at an approximately 2.5 times order of magnitude among both techniques is a possible constraint that the results in the lab compared to the industrial scale are sensitive to discriminate some pores, linked by the quality of the response of the experimental set up dependent upon each technique.

In any case, it seems that analysis of the complete cartridge by the LLDP-based method here proposed did not result in any loss of accuracy and can be used regularly without the need to destroy the cartridge to get smaller coupons.

Finally, the influence of the LLDP analysis on the filters has been tested through the measurement of the clean water permeability. This determination was done, for all 6 cartridges analyzed, before and after the LLDP analysis. Results are presented in the last column of Table 1 and for all cartridges, the differences in water permeability between values obtained before and after analysis were always lower than 5%. This assures the filters have not suffered performance loss due to the LLDP process itself.

## 5. Conclusions

Six MF membrane cartridges for water filtration were analyzed using a precise, accurate, and fast automated CFP commercial device. The device and measurement procedure was adapted to characterize high membrane area modules as those here intended.

The obtained results are very interesting, showing a nice agreement between different runs for both techniques and also a very reasonable agreement was found when comparing GLDP lab-scale (destructive method) with LLDP industrial-scale (non-destructive method) outputs.

The relatively small differences found between both techniques can be considered acceptable as far as membrane manufacturing has always a certain degree of variability since not all variables involved can be totally controlled and very small changes in environmental conditions during the casting process can result in measurable sample to sample differences.

In any case, all the results presented in this study (either coming from conventional GLDP or adapted LLDP) are consistent with the nominal values announced by the manufacturer for each filter.

Certainly, GLDP results must be considered more reliable which is consistent with the fact this technique has been longer considered the standard procedure for determining the PSD of MF membranes. Nevertheless, the results obtained through this industrial-scale adaptation of LLDP are similar enough to that from GLDP. Moreover, the method proposed allows for non-destructive quality control of the filters that in normal conditions cannot be done without destroying the cartridge or asking the selling company for small pieces of the inner membranes. A final advantage of the proposed method relies on the fact that this quality control can be performed regularly just to control the possible loss of performance in the filters due to fouling or particle clogging.

In conclusion, we can assert that an accurate, non-destructive, LLDP-based industrial-scale technique has been proposed. The technique is able to measure inline the porosity characteristics of MF filter cartridges, leading to results fully consistent with GLDP lab-scale porometry.

## Figures and Tables

**Figure 1 membranes-10-00369-f001:**
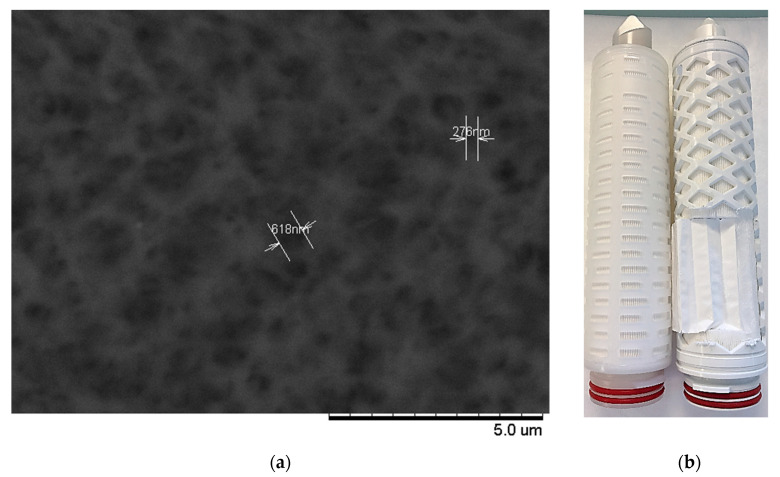
(**a**) Scanning electron microscopy (SEM) images of a PES 0.2 µm membrane showing its porous structure; (**b**) cartridge used for liquid–liquid displacement porosimetry (LLDP) non-destructive analysis and unfolding for gas–liquid displacement porosimetry (GLDP) (autopsy).

**Figure 2 membranes-10-00369-f002:**
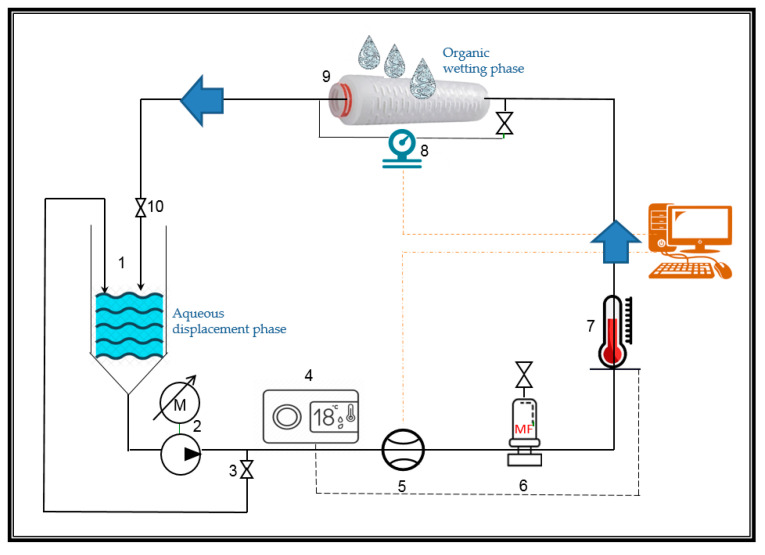
Scheme (**top**) and picture (**bottom**) of the LLDP porosimeter adapted for cartridges testing: **1** reservoir tank, **2** variable frequency pump, **3** bypass valve, **4** thermostatic regulator, **5** flowmeter, **6** pollution filter, **7** temperature sensor, **8** inlet manometer, **9** filter housing, **10** flow regulator valve.

**Figure 3 membranes-10-00369-f003:**
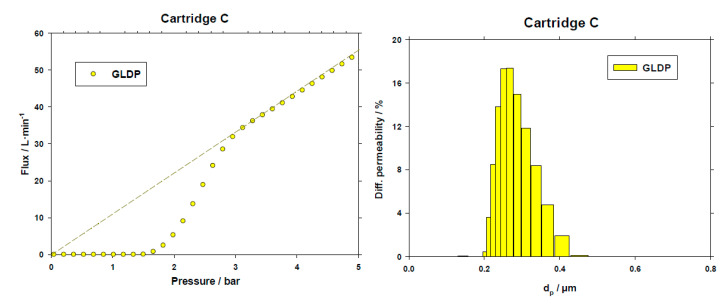
Example of a GLDP porosimetric curve and resulting permeability distribution for a coupon from C cartridge.

**Figure 4 membranes-10-00369-f004:**
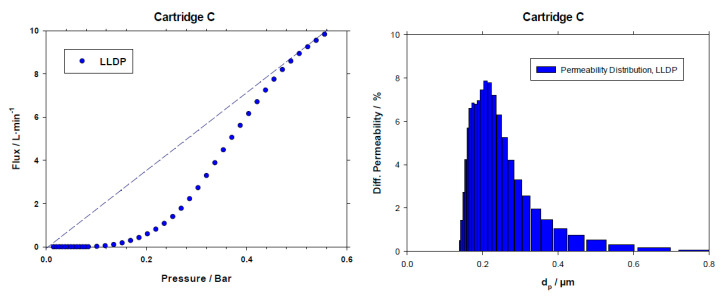
Example of an LLDP porosimetric curve and resulting permeability distribution for the C cartridge.

**Figure 5 membranes-10-00369-f005:**
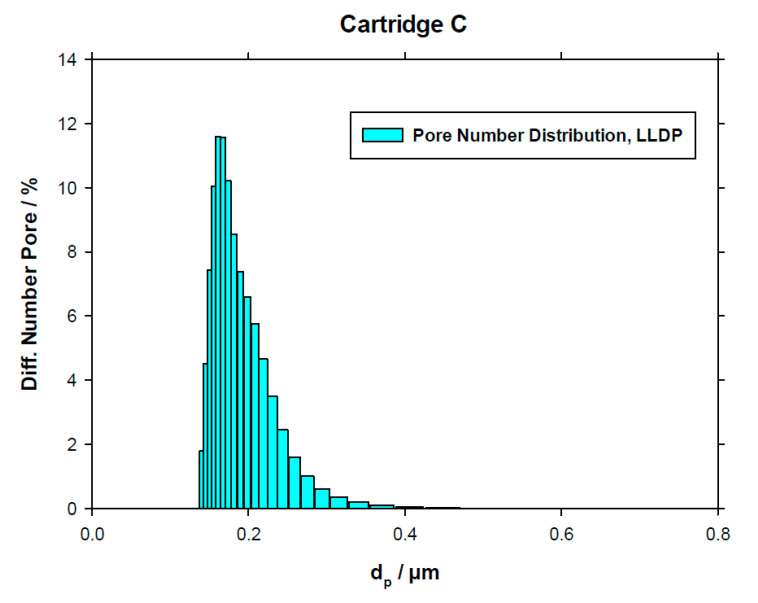
Example of an LLDP porosimetric pore size distribution (PSD) in terms of pore number for the C cartridge.

**Figure 6 membranes-10-00369-f006:**
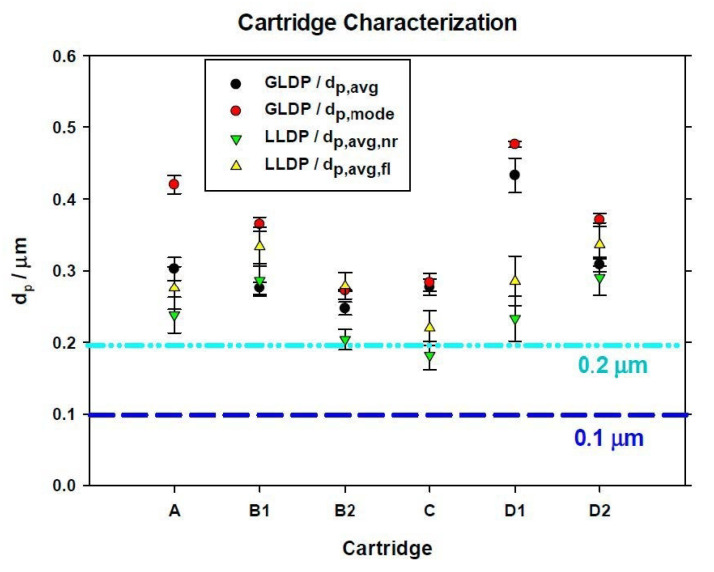
Comparison of porosimetric results for all cartridges studied.

**Table 1 membranes-10-00369-t001:** Characteristics of the six membrane filter cartridges analyzed.

Product	Manufacturer	Material	Nominal Pore Size	Filtration Area /m^2^	Permeability /L·m^−2^·bar^−1^·hr^−1^
A	Company A	Polyethersulfone (PES)	0.2 µm	0.8	~3600
B1	Company B			0.7	~3700
B2				0.8	~2800
C	Company C			0.67	~3400
D1	Company D			0.77	~4600
D2			0.1 µm	0.77	~4600

**Table 2 membranes-10-00369-t002:** Results of the characterization by GLDP and LLDP of the six membrane filter cartridges studied.

Membrane	Nominal Size	GLDP		LLDP	
		*d_p,avg_/*µm	*d_p,mode_/*µm	*d_p,avg_*(*nr*)*/*µm	*d_p,avg_*(*fl*)*/*µm
A	0.2	0.303 ± 0.016	0.420 ± 0.013	0.238 ± 0.025	0.276 ± 0.030
B1		0.276 ± 0.009	0.365 ± 0.010	0.287 ± 0.023	0.334 ± 0.027
B2		0.247 ± 0.009	0.272 ± 0.001	0.204 ± 0.014	0.279 ± 0.019
C		0.277 ± 0.012	0.284 ± 0.013	0.182 ± 0.020	0.220 ± 0.024
D1		0.433 ± 0.024	0.476 ± 0.004	0.233 ± 0.017	0.286 ± 0.022
D2	0.1	0.309 ± 0.010	0.371 ± 0.009	0.291 ± 0.026	0.337 ± 0.030

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
