# Peer review of "Non-Destructive Characterization of Industrial Membrane Cartridges by Using Liquid–Liquid Displacement Porosimetry (LLDP)"

_membranes, 2020, doi:10.3390/membranes10120369_

Round 1
Reviewer 1 Report
This work proposed a method to characterize several microfiltration membrane cartridges by using a Liquid-liquid Displacement Porosimetry (LLDP) for getting their pore size distributions (PSD) and mean pore diameters. The topic is realistic and useful. I think there are some issues that need to be solved before it is considered to be accepted. The suggestions are as below:
- First of all, what kind of advantages does the LLDP have compared to GLDP? According to the authors, applying GLDP to large membrane modules as those concerned in the manuscript is a difficult task, and why can the LLDP overcome this limitation?
- Besides the scheme of the LLDP Porosimeter, the authors may consider to present a photo of the experimental device as well as the testing samples to show the power of the characterization method.
- What is the most prominent advantage of the LLDP compared to GLDP, considering the results of GLDP is more reliable according to the conclusion?
- The authors have mention in the introduction that LLDP has already been proposed to measure the pore size of UF and even NF membranes? What innovation does the work propose? The authors should point that explicitly.
Reviewer 2 Report
- The paper should be re-organized according to the guide for author of this Journal.
- The Abstract should rewrite in order to attract reader's attention.
Round 2
Reviewer 2 Report
The paper had been improved significantly and can be published now.